

# Review on benefits, toxicity, challenges, and future of graphene-based face masks in the prevention of COVID-19 pandemic

Siyanand Kumar Chaudhary[1], Nabina Chaudhary[2], Rahul Chaudhary[2] and Narendra Kumar Chaudhary[1]

[1] Department of Chemistry, Mahendra Morang Adarsh Multiple Campus, Biratnagar (Tribhuvan University), Biratnagar, Province-1, Nepal
[2] Dhaka Central International Medical College, University of Dhaka, Dhaka, Bangladesh

## ABSTRACT

The COVID-19 pandemic caused by SARS-CoV-2 has become a global public health concern. Recently, vaccines have been developed to treat this infectious disease. However, these newly developed vaccines are not widely available and not suitable for all age groups. In such circumstances, it is wise to wear personal protective equipment (PPE) such as masks, gloves, and gowns to better protect against COVID-19. Face masks have long been recommended as a means of preventing respiratory infections. However, inappropriate use of masks may undermine their effectiveness. The antimicrobial and antiviral properties of graphene have sparked interest in the development of medical devices such as face masks, gloves, and gowns with extra filtering ability to curb the effects of the coronaviruses. Their hydrophobicity, nanosize, large surface area, high electrical and thermal conductivities, and virulence are notable features that reduce the transmission of viruses from person to person via respiratory routes. Graphene-enhanced face masks are intended to encourage travelers to wear them at work and during recreational activities. Moreover, graphene can pose health hazards if inhaled during respiration. In this review, we summarize the current status of graphene and its promising applications for combating COVID-19. Additionally, this review aims to explore the quality of this biomaterial and possible suggestions for the better and safer use of graphene structured respirators.

# INTRODUCTION

The current COVID-19 pandemic, caused by the SARS-CoV-2 virus, has become the most debated infectious disease of the 21st century. It poses an unprecedented threat to human health, food habits, travel routes, financial resources, and the work environment (*Buonsenso et al., 2021*; *Guan et al., 2020*; *Huizar, Arena & Laddu, 2020*; *Shaikh, 2021*; *Varma et al., 2021*). According to the WHO's last updated COVID-19 record (by January 11, 2022), 308,458,509 confirmed cases, 5,492,595 confirmed deaths, and 9,138,211,378 vaccine doses administered have been recorded worldwide. Although the number of deaths

Corresponding author
Narendra Kumar Chaudhary,
chem_narendra@yahoo.com

per day has decreased, COVID-19 has not been fully controlled and we are bound to follow precautionary measures to prevent its spread. Many countries have declared complete control over COVID-19, but they are still preparing for a worse situation, as they are more likely to mutate and develop new variants. The early stages of COVID-19 were panic, and we were looking for options to stop its spread. Physical distancing, maintenance of well-ventilated rooms, avoidance of crowds, sanitizing hands, coughing into bent elbows or tissues, restricting travel, and wearing face masks have been strongly recommended by the World Health Organization as preliminary precautions to prevent their spread (*Bazaid et al., 2020*; *Dzisi & Dei, 2020*; *Matuschek et al., 2020*; *Morawska et al., 2020*). New variants of the coronavirus species are still emerging with some novel features that create confusion regarding their control. Recently, a tidal wave of new COVID-19 strains, Omicron (B.1.1.529), has forced the international community to declare a state of emergency. Most frontline healthcare workers, as well as the elderly and immunocompromised individuals, are vulnerable to the COVID-19 pandemic. Even vaccinated individuals are at high risk of immune system impairment. The use of face masks is a promising approach to reduce the spread of respiratory diseases such as 2019-nCoV in affected areas (*Abd-Elsayed & Karri, 2020*; *Howard et al., 2021*; *Stutt et al., 2020*; *World Health Organization, 2020*; *Yan et al., 2021a*; *Yan et al., 2021b*). A face mask is part of a non-pharmaceutical intervention that creates a specific barrier to reduce the transmission of respiratory pathogens. Based on the current scenario of wearing masks as respiratory protection equipment, four types of face masks, such as homemade cloth masks, surgical masks, N95 receptors, and activated carbon masks, are ubiquitous (*Ji et al., 2020*; *Ramírez-Guerrero, 2021*; *Tirupathi et al., 2020*). Despite some shortcomings in the mechanism of action, recent research suggests that medical (surgical masks) and nonmedical (homemade cloth masks, N95 receptors) face masks are very effective in providing a superficial level of protection against SARS-CoV-2 transmission (*Ji et al., 2020*; *Karmacharya et al., 2021*; *Long et al., 2020*; *Sharma, Mishra & Mudgal, 2020*). Therefore, for the defense against the COVID-19 scare, face masks have entered the vast area of the market (*Chua et al., 2020*; *Missoni, Armocida & Formenti, 2021*). The projected inclined figure of the number of masks indicates the need for sufficient raw materials to produce masks in large quantities (*Cumbler et al., 2021*; *Park et al., 2019*; *Worby & Chang, 2020*). The market value of face masks is rapidly increasing, and the desire to offer superior quality filtration in portable respirators has encouraged researchers and the production sectors to explore new materials. Researchers are trying to find a material that can filter air more quickly and efficiently than the materials currently available in face masks (*Das et al., 2020*; *Parlin et al., 2020*; *Shanmugam et al., 2021*).

Natural and synthetic polymer fibers are frequently used as filtering agents in respirators. They are found in various respirators, including surgical face masks and fabric masks (*Akduman, 2021*; *Armentano et al., 2021*). Polypropylene, polyurethane, polycarbonate, and polyethylene are examples of synthetic polymers used as filtering agents in respirators (*Ogbuoji, Zaky & Escobar, 2021*; *Pu et al., 2018*). In addition, cellulose, micro cellulose, and nanocellulose are natural fibers with different morphologies that are used to form a filtering membrane of face masks to prevent the entry of pathogens into the respiratory tract (*Garcia et al., 2021*). Owing to their surface and bulk properties, such as water

permeability, hydrophilicity, and resistance to biofouling, cellulose nanomaterials have demonstrated promising prospects for the development of membranes for viral filtration applications (*Junter & Lebrun, 2017*; *Trache et al., 2020*; *Wang et al., 2013*). The large pore size of non-woven natural and synthetic polymer-based face masks, compared to the sizes of pathogens, particles, respiratory droplets, and aerosols, is the principal cause of the reduction in the mechanical process of air filtration (*Jung et al., 2014*; *Salter, 2021*; *Santos et al., 2020*; *Tcharkhtchi et al., 2021*). Therefore, finer materials with pores smaller than those of pathogens, particulates, and droplets are essential, and there is a demand for creating excellent respirators.

Graphene ensures better respiration when embedded in air-filtering membranes of respirators (*Gope, Gope & Gope, 2021*; *Goswami et al., 2021*). It has a two-dimensional structure in which $sp^2$-hybridized carbons are arranged hexagonally in a honeycomb lattice (Fig. 1). Because of its single layer of carbon atoms, it has a very high surface-to-mass ratio. A key feature of graphene is its large surface area, which makes it suitable for interfacial interactions (*Innocenzi & Stagi, 2020*; *Nguyen et al., 2019*; *Pranno et al., 2020*; *Reina et al., 2021*; *Zou et al., 2016*). The high electrical conductivity, large surface area, photocatalytic activity, and hydrophobic nature of graphene have attracted the interest of many researchers for the design of high-quality respirators (*Cheng et al., 2017*; *Kasbe et al., 2021*; *Maqbool et al., 2021*; *Stanford et al., 2019*). As they are extremely hydrophobic and microporous, they do not allow aerosols, water droplets, particles, or pathogens to remain in the outer layer of the respirators for long periods (Fig. 2). In addition, graphene-derived nanomaterials such as graphene oxide (GO) and reduced graphene oxide (RGO) contain -COOH, -OH, -CONH$_2$, and -C-OH moieties (Fig. 1), which effectively interact with bacterial and viral cell membranes and rupture their outer envelopes. The pore size of the graphene membrane (5.7–25.2 Å) is smaller than the virus size (0.05–0.14 µm) (*Jayaweera et al., 2020*). Therefore, it has a greater tendency to serve as a selectively permeable membrane to separate the pernicious SARS-CoV-2 (*Castelletto & Boretti, 2021*; *Liu, Jin & Xu, 2015*). It is a light-sensitive material that can absorb 2.3% of the incident visible light (*Li et al., 2019*). The amount of light absorbed can increase the temperature of graphene by more than 56 °C, which is sufficient to expel SARS-CoV-2 from the outer surface of graphene-coated face masks within 30 min (*Yang & Wang, 2020*). Exposure of functionalized graphene face masks to sunlight for 10 min can increase the antibacterial efficiency by 8% to 99.99% (*Huang et al., 2020*). Additionally, the sanitization and washing of graphene-loaded face masks are less tedious than those of other mask types. Furthermore, the eco-friendly and reusable features of graphene-based surgical and nonsurgical masks appear to be popular among users.

Several graphene functionalized materials have shown antibacterial properties, and their effectiveness in killing bacteria is encouraging (*Krishnamoorthy et al., 2012*; *Li et al., 2014*; *Liu et al., 2011a*; *Liu et al., 2011b*; *Perreault et al., 2015*; *Yang et al., 2017*; *Zhang & Tremblay, 2020*; *Zhao et al., 2013*). Despite the enormous potential of graphene in a wide variety of biomedical applications, such as drug delivery, chemotherapeutic agents, electron transport systems, enzyme-induction, and bone defect repair (*Abbasi et al., 2016*; *Behbudi, 2020*; *Dhinakaran et al., 2020*; *Du et al., 2020*; *Kumar & Chatterjee, 2016*;

**[a]**

**[b]**

**[c]**

**Figure 1   The structure of the graphene family.** (A) Pristine graphene (PG) does not possess functional groups. (B) Graphene oxide (GO) contains several functional groups for binding. (C) Reduced graphene oxide (rGO) possesses few active functional moieties for binding to composite materials.

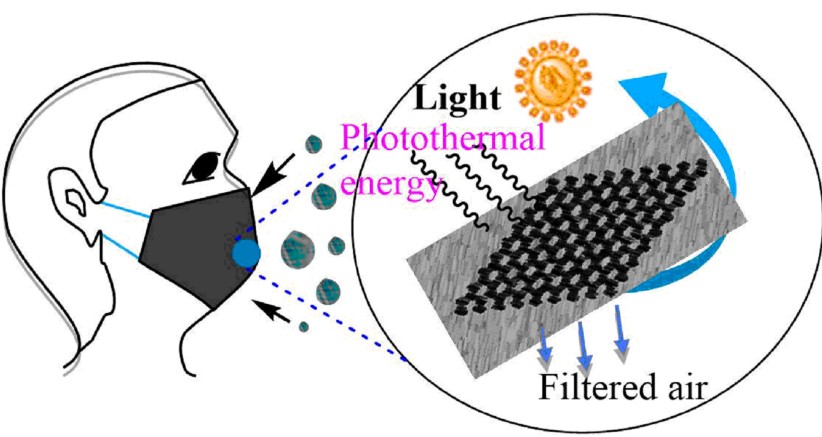

**Figure 2   Graphene face mask showing filtration of aerosolized particles in the presence of sunlight.**

*Perini et al., 2020*), only a few studies have addressed its application in virus filtering membranes (*Barbhuiya et al., 2020*; *Matharu et al., 2020*; *Musico et al., 2014*). An overview of face masks made from graphene and its derivatives is presented in this paper, highlighting their antimicrobial characteristics to reduce the spread of infectious and fatal diseases such

as COVID-19. Moreover, this review addresses the benefits, challenges, and future outlook of masks functionalized with graphene and its derivatives.

## SURVEY METHODOLOGY

The literature referenced in this study was systematically reviewed and searched using PubMed, Google Scholar, and various internet websites. We set no time limits for the search. A manual search was performed to collect appropriate literature. This search was conducted based on title, author name, journal scope, and year of publication. The keywords used to search the literature were ''face mask and types'' or ''face mask and graphene'' or ''face mask and COVID-19''.

### Air filtration by graphene face mask

Face masks are considered safety gear since they protect the respiratory system from airborne droplets and particles. Developing a mask with adequate comfort and high efficacy for removing bio-aerosols, airborne particles, microorganisms, and the particulate matter requires the selection of novel materials and an understanding of the filtering mechanisms in various environments. The face mask efficiency can be affected by many factors, including the inherent properties of the material, chemical composition of the filter, fiber thickness in the filter membrane, and packaging density (*Konda et al., 2020*; *Leung et al., 2020*). Moreover, many external factors, such as gravitational force, air velocity, electrostatic charge, frequency of respiration, relative humidity, temperature, loading time, and particle interception, contribute to disturbances in air filtration (*Rengasamy et al., 2018*; *Tcharkhtchi et al., 2021*). Scientists have used natural and synthetic polymers for decades to make standard-grade face masks. Owing to the failure of polymer-based face masks to meet the standard values and norms, attempts have been made to replace them with graphene, graphene oxide, reduced graphene oxide, and metal-based nanoparticles.

Graphene is a hydrophobic material that is used in face masks to quickly remove respiratory droplets (*Deng et al., 2021*). Aerosolized particles ranging in size from 1 to 10 mm were trapped inside the pores of the outer layer of a traditional face mask. Graphene-based face masks have a unique filtration system that prevents water droplets from attaching to their surfaces and remaining there for an extended period. To verify the filtering efficiency of the 3D printed face mask, *Goswami et al. (2021)* used functionalized graphene to fabricate filtering membranes of 20, 10, and 3 mm made from polypropylene. Aerosol particles containing viruses and bacteria were allowed to pass through three layers of the membrane: the outer and inner layers without graphene, and the middle layer with graphene, and they found that bacteria and viruses were trapped more in the middle layer. They explained the air filtration process in three ways, based on the materials used and the results obtained. Initially, they believed that graphene had a sharp edge similar to a nano blade that would tear apart the virus' spike protein. Additionally, they hypothesized that electrostatic interactions with living particles may have played a significant role in trapping them, and third, they suggested that the pore size and hydrophobicity of functionalized graphene may have resulted in superior filtration.

A key part of the material is the functional groups of activated graphene oxide, which improves the filtration process (*Chung et al., 2021*; *Rhazouani et al., 2020*; *Song et al., 2015*). In contrast to pristine graphene, graphene oxide and reduced graphene oxide interact more rapidly with the outer lipoprotein layer of bacteria and viruses as they pass through the different layers of the face mask. As bacteria interact rapidly with graphene oxide, other factors such as breathing speed, germ size, and droplet diffusion do not influence how well it filters the air. Furthermore, the addition of graphene oxide to the filtering membrane of the face mask increases the charge density and has a stronger electrostatic effect on microorganisms. *Donskyi et al. (2019)* developed a platform to investigate the electrostatic interactions between functionalized graphene and herpes simplex virus type 1 (HSV-1). Their study showed that electrostatic forces were the primary driving force behind the virus trapping. Therefore, graphene-based face masks are believed to provide the maximum protection against disease-causing microbiological particles by acting as excellent filters.

*Pal et al. (2021)* measured the filtration efficiency of laser-induced graphene face masks in light. Light is the main source of photothermal energy that is used to heat the filtering membrane of the face mask. Exposure of face masks to light with a wavelength of 1,085 nm for 15–20 min improves filtering efficiency by 99.98%. Therefore, the filtration of air depends not only on the materials used in the membrane, such as graphene nanoparticles but also on the light source used. *Lin et al. (2021)* evaluated the performance of a face mask using the hydrophobicity of graphene material. Graphene nanosheet-embedded carbon face masks are believed to exhibit excellent performance in air filtration owing to the hydrophobic nature of graphene.

## Recent advances in graphene-based face masks and benefits

Recently, interest in graphene-derived 2D nanomaterials such as nanoporous graphene, graphene oxide (GO), reduced graphene oxide (RGO), graphene quantum dots (GQDs), and other graphene-derived materials has increased significantly (*Catania et al., 2021*; *Jiang et al., 2021*; *Saleh & Fadillah, 2019*; *Yan et al., 2021a*; *Yan et al., 2021b*). These materials are particularly well suited for various ionic sieves, molecular separation, desalination, gas-phase separation, dialysis, hemofiltration, ultra-filtration, water sterilization, sensors, protein separation, viral extraction, and other biomedical applications (*Ali et al., 2020*; *Ali et al., 2019*; *Thebo et al., 2018*). Studies have also demonstrated the synergistic effect of graphene when used in air filtration masks. However, there are very few commercially available graphene-functionalized nanofiber respirators. Face masks would be effective and acceptable only if aerosolized particles are promptly prevented from entering the respiratory tract.

To ensure superior air filtration in respiratory devices, designers must be knowledgeable about the best practices for seeding, polishing, coating, and synthesizing graphene nanoparticles on the nanofibers. Of the many approaches used to seed graphene nanoparticles on a polymer matrix, electrospinning is one of the most versatile and viable. Electrospinning is used to disperse nanoparticles into ultrafine nanofibers with minimal diameters and produce very fine fibers. This is a very reliable method for storing electrical charges in membranes to improve their air filtering performance (*Bortolassi et al.,*

*2019*; *Li et al., 2018*; *Zhang et al., 2020*). *Goswami et al. (2021)* fabricated a 3D-printed face mask from polylactic acid and coated it with functionalized graphene ink, and measured the virus arresting, capturing, and filtering efficiencies of the mask. This result was exciting and supported the use of functionalized graphene as a filtering and antibacterial agent.

Graphene, a material with antimicrobial and antiviral properties, has increased the interest of the scientific community in investigating its use in preventive measures, detection, and diagnosis of COVID-19 (*Damiati et al., 2021*; *Mojsoska et al., 2021*; *Payandehpeyman et al., 2021*; *Pinals et al., 2021*; *Raval et al., 2020*; *Torrente-Rodríguez et al., 2020*). Graphene-based face masks are newly developed biocompatible medicinal weapons that seem worthy of facing the COVID-19 pandemic. What makes the graphene face masks extraordinary compared to others is shown in Fig. 3. The antibacterial, antistatic, large surface area, sharp edge, photosensitivity, and electrical superconducting nature of graphene nanomaterials are well suited for designing a shielding membrane in face masks and provide a lot of benefits to face mask holders (Fig. 4). Private companies, BonBouton have developed reusable, non-disposable, electrothermally and photothermally self-sterilizing, and rechargeable graphene face masks with a functional graphene-infused film that can quickly block viruses from getting inside the respiratory trachea (*Maqbool et al., 2021*). To reduce the pain stacked and mourning situation of the COVID-19 pandemic, ZEN Graphene Solution Ltd. and Graphene Composite Ltd. (GC) have also developed a graphene-based composite ink for manufacturing mouth-nose-covering devices and other personal protective equipment (*Gope, Gope & Gope, 2021*). Using silver and graphene nanoparticle composite inks, they have modified the working mechanism of earlier cotton- and textile-based face masks, which can now efficiently disable SARS-CoV-2 and influenza A and B virus strains (*Chaudhary et al., 2021*). Planar TECH and IDEATI have also manufactured graphene-coated cotton fabric 2 AM face masks (*Gope, Gope & Gope, 2021*; *Maqbool et al., 2021*). Moreover, Goswami and his coworkers have successfully developed and tested a graphene-based 3D-printed facial protection device active against the SARS-CoV-2 virus (*Goswami et al., 2021*). From the imperial findings, they have inferred that the working principle of the face mask to filter the virus is quite interesting. Likewise, Directa Plus has also used thin atom allotropes of carbon in face masks extracted from graphite to reduce the spread of viral diseases. A skin-tested hypoallergenic G+ mask can offer consumers a range of benefits to protect themselves and others from viral infections (*Lea, 2020*).

*Zhong et al. (2020)* reported a dual-mode laser fabrication technique for depositing graphene onto temperature-sensitive surgical masks. They found that functionalized graphene face masks with super hydrophobic surfaces offered enhanced protection against coronaviruses from respiratory droplets. Furthermore, *Shan et al. (2020)* revealed that electrothermal graphene-modified masks (GMMs) exhibited good performance in preventing particulate matter and viruses from entering the nose. The findings also showed that GMM is far more efficient than the photothermal face mask for purging breathing air. The most attractive features of the graphene-engineered face masks are their fast-charging capabilities, ability to maintain a temperature of 80 °C by supplying 3V power for 5 h, and reusability and biodegradability. Currently, researchers are well-versed in the many

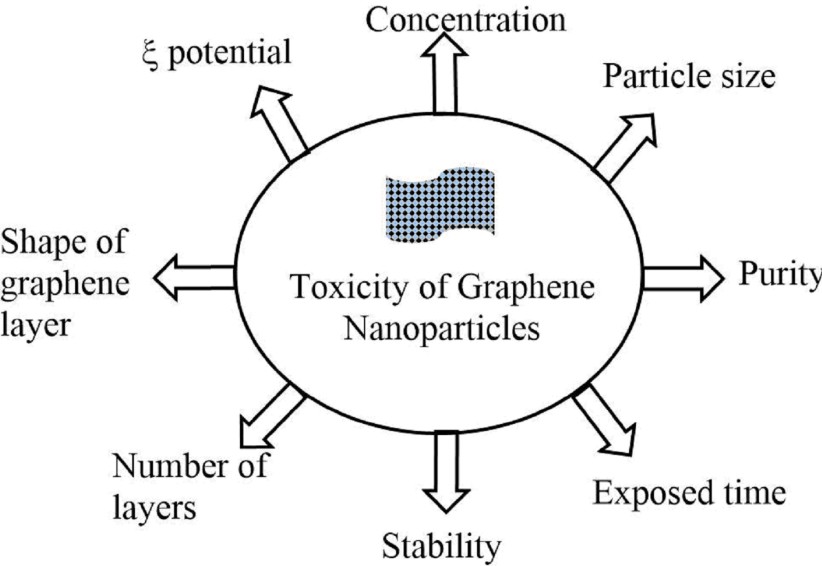

**Figure 3** An overview of various physico-chemical parameters affecting the toxicity of graphene nano-materials.

valuable characteristics of graphene. Some researchers have used the unique properties of graphene, such as photosensitivity, to create photothermally self-sterilizing and reusable face masks to reduce the financial and environmental costs associated with the subsequent use of disposable face masks. Graphene nanosheet-embedded carbon (GNEC) film face masks are perfect examples of ways to meet the necessary conditions for improving air filtration quality (*Lin et al., 2021*). Another private company, Medisevo, claimed that their graphene-based face mask developed by them could filter 98% of COVID-19 particles. Medisevo evaluated graphene face masks using the medical face mask standards from the American Society for Testing and Materials (*Sandle, 2021*). LIGC Technology has developed a face mask called the "Guardian G-Volt" made from laser-induced microporous graphene. The originality of this facemask is that it maintains an electrical charge to kill the microbes trapped in the filter, effectively blocking 99% of contaminants with sizes greater than 0.3 μm. The graphene face mask G1 Wonder, developed by Nanometric Materials Pvt. Ltd. with a composite membrane of graphene and silver nanoparticles, can kill 99% of bacteria and viruses (*Moore, 2021*). Laboratory tests have shown that graphene-silver, a composite membrane made from a collection of microscopic razor-sharp blades of graphene with a high electrically charged potential, has the power to break, open, and destroy bacterial and viral cells. Therefore, masks can prevent COVID-19 transmission by preventing the virus from passing through the mask membrane. Table 1 reports the different types of graphene face masks and their empirically justified outstanding air-filtering features. In addition, graphene face masks can act as antimicrobial agents even after 10 washes (*Ray & Bandyopadhyay, 2021*). This is a unique features of this mask, that is not present in many trivial face masks. Other notable features of the graphene-enhanced face masks are shown in Fig. 5. Consequently, the users have numerous advantages. Laser-induced graphene face

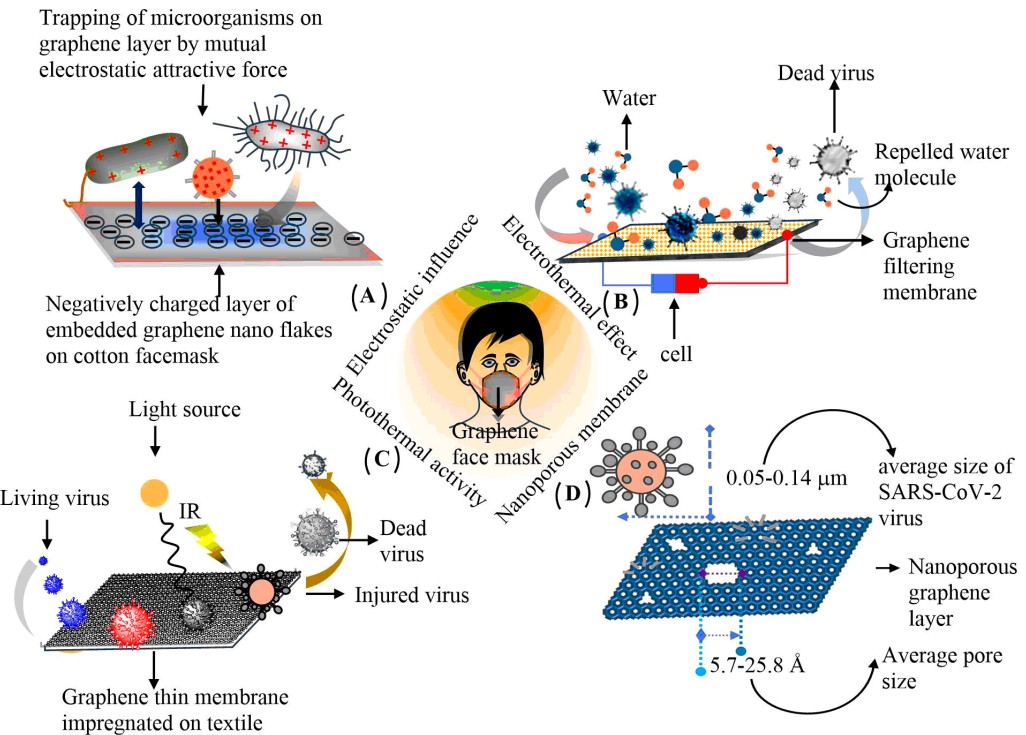

**Figure 4** **An illustration showing the filtering efficiency of graphene-based face masks.** (A) Graphene materials trap microorganisms by the electrostatic force of attraction. (B) Electrothermally active graphene face masks inactivate harmful viruses more effectively. (C) A cloth-mounted graphene membrane can destroy the protein envelope of the virus in presence of IR radiation. (D) The average diameter of graphene nanopores is smaller than the size of the SARS-CoV-2 virus.

masks have several excellent physical and chemical mechanisms for fighting SARS-CoV-2 infections (Fig. 4) (*Pal et al., 2021*).

Graphene-modified face masks can be sterilized using photothermal or electrothermal energy, which is a remarkable feature, not found in surgical masks like N95, FFP3, P100, KN 95, and N99 masks. Owing to its high electrostatic charge retention capacity, it is several times more effective in filtering air than other popular face masks. Furthermore, graphene face masks minimize the use of non-biodegradable materials, thereby ensuring a clean and pollution-free environment. The free electrons of graphene nanoparticles are used to trap positively charged bacteria and viruses in face masks (*De Maio et al., 2021*; *Seifi & Reza Kamali, 2021*). Their high electrical conductivity and the flow of electrons from graphene-based materials cause oxidative stress in bacteria and viruses. As a result, protein denaturation and the destruction of cellular components occur rapidly. Furthermore, the mutual Van der Waals attraction between the embedded graphene nanomaterials and the germs in the droplets prevents the spread of microorganisms from person to person via air roots (*Kumar et al., 2019*). The electrical conductivity of graphene-derived nanomaterials also supports biosensors in detecting, trapping, inactivating, and preventing viruses from spreading. Several studies have emphasized the importance of graphene in the manufacture

**Table 1  Features of different types of graphene face masks.**

| S.N. | List of grapheme facemask | Features | Ref. |
|------|---------------------------|----------|------|
| 1. | Graphene nano sheet-embedded carbon (GNEC) facemask | The mask has an excellent hydrophobic property, incredible bacterial filtering efficiency, and prominent photo-sterilized performance. Masks have great potential to work against the spread of the COVID-19 pandemic. | *Lin et al. (2021)* |
| 2. | Graphene mounted 3D printed facial mask | The bacterial filtration efficiency of the mask is 98.2% and the breathing resistance is 1.10 mbar. Transmission of the SARS-CoV-2 virus through graphene filters was not reported. | *Goswami et al. (2021)* |
| 3. | Flextrapower graphene mask | The mask is carefully made and is safe to wear. The mask follows unique and sophisticated hydrophobic nanotechnology in which virus aerosolized droplets are unable to remain on the exposed layer for long periods. | *Nacinopa (2020)* |
| 4. | Laser-Induced graphene mask | Most of the bacteria remain alive even after the mask is exposed to sunlight for 8 h. The mask has shown superior antibacterial action and can be enhanced by photothermal energy. | *Huang et al. (2020)* |
| 5. | Endothermal mask | It is very easy to raise the temperature of the face mask above 80 °C by supplying 3V of energy. The mask filters the air at this temperature by killing all known types of bacteria and viruses. It preserves high particulate matter efficiency and is reusable. | *Shan et al. (2020)* |
| 6. | Graphene oxide-based rechargeable respiratory mask | The mask comes at a very low price (~$1/mask). Its rechargeability and filtering efficiency are more than 95%. The electrostatic charge retention capacity is very high (~1nC/cm2). Even in high humid conditions, the mask recharges very quickly. | *Figerez et al. (2020)* |
| 7. | G+ masks | The G+ mask has been certified to the European standard EN 14683 as an excellent air filtering biocompatible device. It is naturally bacteriostatic and hypoallergenic. The ability to filter while breathing with this mask is very high. It is reusable, washable and the filtering membrane is replaceable. | *Bhattacharjee et al. (2019)* |
| 8. | G1 wonder mask | Graphene-silver nanomaterials are used to design the filtering membrane. This increases the filtering efficiency of the G1 Wonder Mask. The mask is reusable, washable, breathable, and eco-friendly. It can kill 99% of bacteria and viruses in just one second, and also prevent the volatile organic compound from entering inside respiratory organs. | *Moore (2021)* |
| 9. | Graphene masks | The mask is super-hydrophobic due to the embedded graphene nanoparticles. Exposure to sunlight can raise its temperature to 80 °C which is enough to kill bacteria and viruses. In this mask, monolayered nanographene particles are deposited on a non-woven surface at low melting temperatures by a laser-induced forward transfer method. | *Zhong et al. (2020)* |
| 10. | 2 AM graphene enhanced facemask | This type of face mask consists of three-layered materials such as graphene (outer layer), polyester (middle layer), and 100% cotton (inner layer). Graphene material is anti-static, dust repellent, and filters out PM2.5 airborne particulates. It is washable and also bacteria resistant. The even distribution of heat energy from graphene provides additional comfort to the users of this mask. | *Maqbool et al. (2021)* |

**Table 1** (*continued*)

| S.N. | List of grapheme facemask | Features | Ref. |
|---|---|---|---|
| 11. | Graphene facemask | This is an antibacterial face mask, and the antibacterial properties remain the same even after washing the mask 10 times. It comes in a variety of shapes and sizes, and the mask is flexible as well. In addition, users will feel comfortable while breathing in hot or cold weather. | *Kilgannon (2020)* |
| 12. | Medieval facemask with graphene | The mask is comfortable, durable, and wearable without fogging glasses. Antibacterial and hypoallergenic ingredients have been used in this mask. From the inner side, the organic lining has been retained for soft and comfortable wear. | *Sandle (2021)* |
| 13. | Guardian G-Volt masks | The mask is rechargeable and can be sterilized and reusable. It shows antimicrobial properties and repels the microorganism by attachment to the exposed surface of the face mask. | *Pullangott et al. (2021)* |
| 14. | G/GO-functionalized polyurethane or cotton facemask | The G/GO nanoparticles functionalized cotton face mask has significantly enhanced protection against the SARS-CoV-2 virus. It has shown antibacterial properties when the material was tested against E. coli. | *De Maio et al. (2021)* |
| 15. | Anti-COVID laser-induced graphene mask | The mask has superhydrophobic and reusable properties. Sunlight empowers the sterilization of facemask. Exposure of a mask to sunlight can increase its temperature by more than 80 °C. | *Pal et al. (2021)* |

of sensors as a virus detection agent in clinical settings (*Bardhan, Jansen & Belcher, 2021*; *Jung et al., 2010*; *Liu et al., 2011a*; *Liu et al., 2011b*; *Seo et al., 2020*). Furthermore, good electrical conductivity boosts charging speed. Thermal energy also plays an important role in the denaturation of the S-protein and the inactivation of the virus. Thermal exposure of the virus to temperatures of 75 °C for 3 min, 65 °C for 5 min, and 60 °C for 20 min left no option for its survival.

## Limitations, challenges, and the risk of using graphene face masks

As graphene-functionalized face masks are gaining popularity owing to the dire situation of the COVID-19 pandemic, the attention of many researchers has turned to safety awareness, exploring the potential dangers of graphene-seeded respiratory masks (*Fadeel et al., 2018*). Many analysts believe that such face masks can significantly disrupt the spread of the COVID-19 pandemic by breaking the chain of virus transmission from one infected person to another. A mask containing graphene and its derivatives may cause long-term adverse effects on the user's skin, vital respiratory, circulatory, excretory, and digestive organs (*Arvidsson, Molander & Sandén, 2013*). The lungs may be damaged when grapheme particles reach them after breathing through a mask made of graphene filters. A scientific report has shown that inhalation of graphene nanoparticles can pose serious unexpected risks to lung tissues and blood circulation (Fig. 6) (*Ingle et al., 2013*; *Schinwald et al., 2012*; *Wang et al., 2016*). Based on empirical results, the ability of graphene and its derivatives to inhibit the activation of living cells and the circulation of blood is dose and particle size-dependent (*Mukherjee, Kostarelos & Fadeel, 2018*). An experimental study conducted

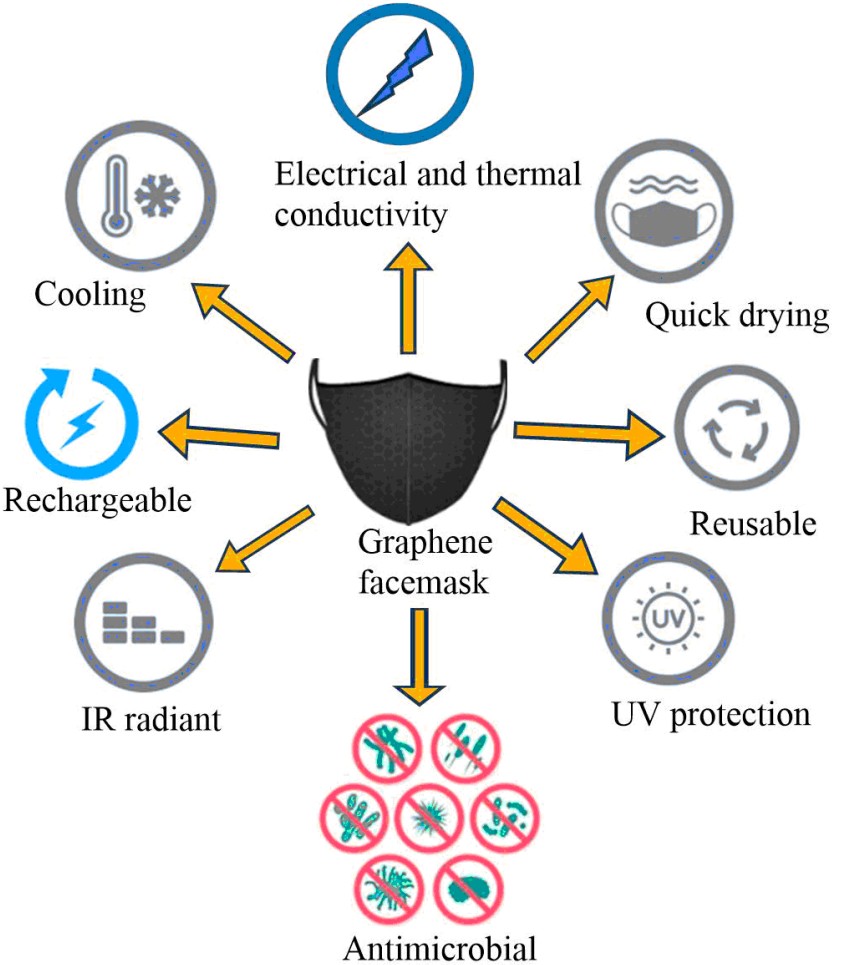

**Figure 5** **Schematic illustration of graphene engineered face mask showing the physical and mechanical features.**

by *Zhang et al. (2010)* found a decrease in metabolic activity with 0.1 mg/L graphene in mice, but no effect with a 0.01 mg/L concentration. Mice exposed to aerosolized graphene environments develop lung damage, inflammation, lung granulomas, pulmonary edema, and persistent lung injury (*Das et al., 2020*; *Parlin et al., 2020*). Inhalation of graphene in mice caused more severe lung injury than asbestos inhalation in humans. Although it helps reduce the number of COVID-19 patients, the effects of graphene on the cellular level of living organisms are considered hazardous rather than benign. The toxicity of graphene nanomaterials at the cellular level is summarized in Table 2.

*Li et al. (2013)* exposed the skin of some selected people *in vitro* to a suspended graphene atmosphere and found that the plasma membrane invasion of primary human keratinocytes is due to the aggregation of few-layer graphene-derived material on the dermal layer. Therefore, it is thought that this may also be a possible cause of keratinocytes in people who wear graphene, graphene oxide, or reduced graphene oxide integrated respirators.

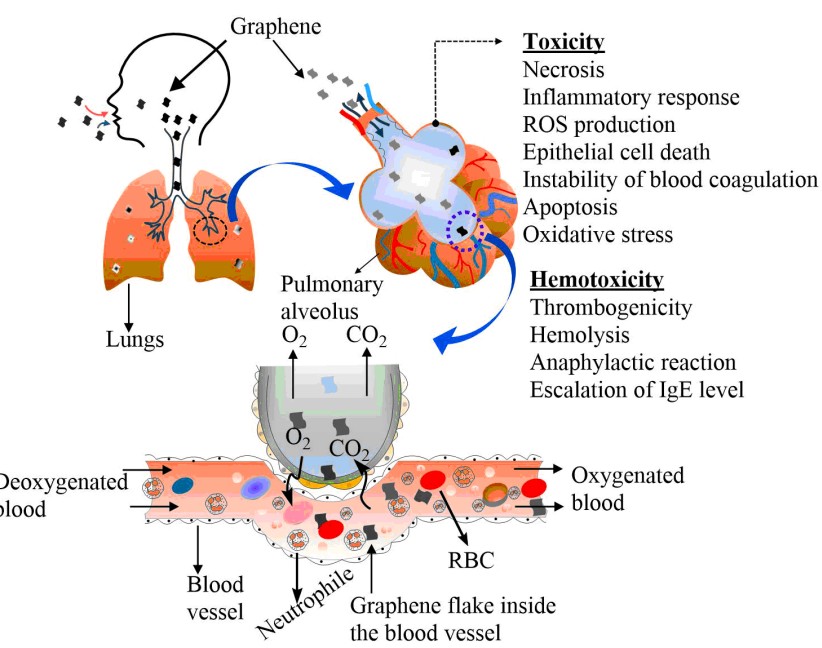

**Figure 6** **Proposed oral and nasal routes of entering graphene nanoflakes in lungs and blood vessels.** The graphene nanoparticles reach inside the blood vessel by rupturing the inner lining of alveolar epithelial cells and initiate the problem of hemotoxicity, thrombogenicity, anaphylactic reaction, and escalation of IgE level.

If masks structured with graphene and its derivatives are worn continuously for a long time, the possibility of nanoparticle adsorption into the skin of the mask covering the area increases (*Schinwald et al., 2012*). Thus, long-term use of graphene has the potential to cause skin allergies and damage epidermal cells. The effects of different GO panels (lateral dimension: 871–1,678 nm; thickness: 1 nm–10 nm) and graphene sheets (average lateral dimensions: 4,312 nm; thickness: up to 10 nm) on fibroblast skin were evaluated by *Liao et al. (2011)*. Their results demonstrated that graphene is more invasive and lethal to dermal cells than graphene oxide because it tends to aggregate within the cells. The MTT method, widely used to assess the toxicity of nanomaterials in cell culture has demonstrated that the metabolic activity of PC12 cells decreases in a concentration-dependent manner after 24 h of exposure to graphene nanoparticles. Along with promising results, graphene nanoparticles are highly cytotoxic, and the cytotoxicity of nanoparticles varies based on particle dimensions. Furthermore, the toxicity of graphene-based nanoparticles became more apparent in studies of histopathological changes elicited after exposing mice to graphene aerosol environments for five days (6 h per day). Histopathological inspection successfully explained the rupture of macrophage cells in the presence of 3.05 and 10.1 mg/m$^3$ graphene (*Ma-Hock et al., 2013*).

As a source of material for manufacturing of graphene impregnated face masks, graphene nanoparticles may be risky, unrealistic, and perilous (*Farahani et al., 2020*; *Palmieri & Papi, 2020*; *Zhou & Gao, 2014*). Over the past few decades, numerous scientific investigations have alerted investors, researchers, material scientists, chemists, pharmacists,

Chaudhary et al. (2022), *PeerJ Materials Science*, DOI 10.7717/peerj-matsci.20

**Table 2  Toxicity of graphene nanomaterial at a cellular level.**

| Nanoparticle | ξ pot | Size of Nanoparticle | Cells | Dose (µ g/ml, µ g/Kg) | Time after administration/ Incubation | Toxicity assessment | Result/effect | Ref. |
|---|---|---|---|---|---|---|---|---|
| Pristine graphene (PG) | −6.12 mV | Surface diameter ranges from 350 nm to 6 µ m | HS-5 cells (bone marrow/ stroma) | (5–100) µ g/ml. | 24 h | The metabolic rate of HS-5 cells and Intracellular ROS level. | The result showed high interaction of cells with pristine graphene and very fast agglomeration of PG nanoparticles on the cell surface. The viability of cells began to decline after 20–100 µ g/mL. Additionally, the level of ROS was increased to 20 µ g/mL at the higher concentration of PG. | *Jaworski et al. (2021)* |
| Graphene Nanoparticles (GNPs) | No information | No information | Lung cancer cells (SKMES-1, A549). | (5, 50, 250, 500, 1,000) µ g/ml | 24 h | Toxicity of graphene nanopores on lung cancer cells. | The toxicity of graphene was concentration-dependent. This resulted in late apoptosis at a concentration of >250 g/ml. At low concentrations, GNP does not significantly cleave the cell membrane. | *Tabish et al. (2018)* |
| Graphene Sheet (GS) | −37.2 ± 1.6 mV | Hydrodynamic diameter 3,018 ± 36 nm in D.I. water | Human Red Blood Cells (RBC), Human Skin Fibroblast cells (CRL-2522) | (3.125–200) µ g/ml | 24 h | ROS generation, hemolytic effect on RBC, apoptosis of viable cells. | The surface charge of GS is responsible for the hemolytic activity of erythrocytes and is dose-dependent. The comparative result shows a lower hemolytic activity of GS than that of GO. GS generates more reactive oxygen species (ROS) in human skin fibroblast cells and is strongly associated with the cell surface. | *Liao et al. (2011)* |
| Graphene Nanoparticles (GNPs) | No information | Average hydrodynamic diameter 323.3 nm and width in the aqueous suspension 80.05 nm | Epithelial cells of Human lung (A549) | (0.1–1,000) µ g/ml | (24–72) h | Assessment of viability of A549 in GNPs. | The toxicity of GNPs over A549 is concentration and time-dependent. Furthermore, exposure of GNPs on human epithelial cells of the lung for 72 h, is more lethal than exposure for 24–48 h. | *Nasirzadeh et al. (2019)* |
| Pristine graphene | −20.08 mV | Average diameter 172.7 ± 75.6 nm and thickness 2–3 nm | Murine RAW 264.7 macrophages | 20 µ g/ml | 24 h | Quantification of cytokines and chemokines. | Exposure of macrophages to PG increases the secretion of Th1/Th2 cytokines by activating the NF-$\kappa$B signaling pathway. | *Zhou et al. (2012)* |

**Table 2** (*continued*)

| Nanoparticle | ξ pot | Size of Nanoparticle | Cells | Dose (μg/ml, μg/Kg) | Time after administration/ Incubation | Toxicity assessment | Result/effect | Ref. |
|---|---|---|---|---|---|---|---|---|
| Pristine graphene | No information | Thickness is 2–3 nm and the size is 500–1,000 nm | Murine RAW 264.7 macrophages | (5–100) μg/ml | 48 h | Triggering of apoptosis in macrophages by pristine graphene | PG induces cytotoxicity by breaking the mitochondrial membrane potential (MMP) and increasing intracellular reactive oxygen species (ROS) levels. It also triggers apoptosis by activation of the mitochondrial pathway. | *Li et al. (2012)* |
| Graphene nanoplatelets (GNPs) | −13 ± 0.5 mV | Average thickness 220.26 nm, average hydrodynamic radius 243.4 ± 1.4 nm | Human colorectal adenocarcinoma cells; Coca-2 and HT29 | (0–100) μg/ml | 24 h | Cytotoxicity assessment | Cell viability remains approximately the same at all concentrations of GNPs and no severe cytotoxic effect of GNPs on Coca-2/HT29 cells has been reported. | *Do et al. (2020)* |
| | | | | (5–50) μg/ml | 24 h | Genotoxic damage, intracellular oxidative stress induction | DNA damage has been detected with a comet assay. Initiation of genotoxic damage and expression of the ROS-related gene were GNPs concentration-dependent. | |
| Pristine graphene (PG) and functionalized graphene (FG) | No information | Average thickness ≈0.4 nm, double-layered | Murine macrophage cells (RAW 264.7) and human primary blood components | (0–75) μg/ml | 3 h | Inflammation analysis, ROS production, and induction of apoptosis | PG induces higher cytokines production than FG. In addition, PG showed greater anti-inflammatory potential than FG. Apoptosis increased when PG has changed to FG by surface functionalization. Furthermore, PG enhances the formation of ROS. | *Sasidharan et al. (2012)* |
| Pristine graphene, Lower Oxygen Graphene (LOG), Higher Oxygen Graphene (HOG) | No information | The particle size of pristine graphene is 349 ± 24 nm, LOG 423 ± 9 nm, and HOG 265 ± 48 nm | PC-12 Cell line | (5–100) μg/ml | 24 h | Cell viability, cell toxicity, dispersion in the cellular membrane, and LDH assay | PG is the most cytotoxic and toxicity decreases with increasing oxygen content. At higher concentrations of PG (50 μg and 100 μg), the metabolic activity of viable cells has significantly reduced. HOG and GO showed almost identical cytotoxic results that were two-fold less than LOG. The induction of LDH secretion is due to the breakdown of the cell membrane by graphene nanoparticles. | *Majeed et al. (2017)* |
| | | | | (0.5–5) μg/ml | 2 h | Quantification of Reactive Oxygen Species (ROS) | Oxidized graphene generates higher ROS levels than PG and the aggregation of pristine on the cell membrane facilitates the formation of ROS. | |

and mask producers to the risks associated with using graphene and its derivatives for multidisciplinary purposes. Several studies have highlighted the harmful effects of graphene and its products on the endocrine, reproductive, immune, nervous, gastrointestinal, and other physiological systems of animals, including humans (*Kucki et al., 2017*; *Orecchioni et al., 2017*; *Rajakumari et al., 2020*; *Ramal-Sanchez et al., 2021*; *Shin et al., 2015*). The negative impact of nanographene materials on aquatic, marine, and terrestrial animals and plants suggests that graphene toxicity depends significantly on its concentration and particle size. Some studies have revealed the carcinogenic activities of graphene-based materials (*Banerjee, 2016*). The toxicity of graphene was also influenced by other physicochemical parameters, as shown in Fig. 5. Exposure of graphene oxide at varying concentrations to the protozoan *Euglena gracilis* further demonstrated the toxic effect of graphene oxide in the aquatic environment. Hu et al. discovered that when *Euglena gracilis* was exposed to graphene oxide at a concentration of 2.5 mg $L^{-1}$ for 96 h, it had devastating effects. This concentration increases the level of malondialdehyde, which reduces the growth rate of *E. gracilis* and causes oxidative stress (*Hu et al., 2015*). Additionally, some research has revealed the perilous consequences of graphene-based materials on the marine environment. An experiment with *Artemia salina* showed that the availability of graphene oxide in water at a concentration of 1 mg $ml^{-1}$ affects the swimming behavior and survival of its larvae (*Lu et al., 2018*). Graphene derivatives, such as graphene oxide (GO), quantum dot particles, and reduced graphene oxide (RGO), have also been found to severely affect the metabolic activity, photosynthesis, germination, seedling, growth rate, and flowering of plants (*Jastrzębska, Kurtycz & Olszyna, 2012*; *Xu et al., 2020*; *Zhang et al., 2015*; *Zhang et al., 2021*). Consequently, graphene-based face masks are considered an unhealthy practice in the manufacture of medical protective equipment such as goggles, gloves, aprons, shoes, and filter membranes.

According to recent reports, the Canadian government has issued an awareness notice from Health Canada declaring that graphene-based face masks can endanger the health of users (*Cheng et al., 2021*). For a certain month, in Canada, graphene-based face masks were prohibited from being used. Additionally, the director of the French hospital asked people to stop using face masks that included biomass graphene as the elementary material in the filtering membrane until detailed reports of this face mask were obtained. Owing to the lack of comprehensive studies and insufficient evidence to define graphene as a protective material for use in face masks, suggesting people to use graphene engineered face masks for protection purposes would not be appropriate.

## FUTURE OUTLOOK

Graphene and its derived materials improve inappropriate and less efficient personal protective equipment (PPE) containing cotton, silk, chiffon, flannel-based woven, and non-woven fibrous face masks. Graphene is a promising material for enhancing the filtering efficiency of traditional clothing, surgical, non-surgical, and N95 face masks. Recently graphene nanomaterials have been introduced in the production of air-breathing filter membranes to guarantee the superior quality of air purification. Major private companies

developing large-scale production of graphene face masks to meet global demand are First Graphene, Planar Tech, Zen Graphene Solutions, and Graphene Composites (GC). Recently, many graphene-related face masks are still in the testing phase, which is why no statistical information is available regarding the positive progress of graphene-related masks. Owing to insufficient information about the scope, demand, challenges, and response to graphene-related face masks; it is difficult to immediately estimate the pros and cons of graphene-fabricated face masks. We believe that the future of graphene-based face masks remains unclear. The incredible power of this face mask to eliminate respiratory droplets, particulate matter, toxic pollutants, bacteria, viruses, pathogens, and aerosolized microorganisms indicates that graphene face masks could play a game-changing role in the future when people face the upcoming waves of the COVID-19 pandemic. However, the possibility of getting sick due to the inhalation of graphene nanoparticles from integrated graphene face masks poses a serious threat that could jeopardize its future scope. Moreover, the restricted use of graphene masks in some countries due to toxicity and dangerous effects on the wearer's cellular level suggests that the use of this personal protective equipment may be worthwhile to some extent (*Arvidsson et al., 2018*; *Fadeel et al., 2018*). It is believed that the future of graphene-related face masks is difficult to pinpoint and that portraying them as friends or foes is uncertain.

## CONCLUSION AND FUTURE PERSPECTIVES

Graphene is an exciting material that may offer multiple benefits for establishing effective mitigation strategies to improve healthcare services against SARS-CoV-2. It is a suitable nanomaterial for embedding coated clothing in PPE, face masks, and gloves to make medical devices more manageable, and efficiently inhibit the spread of SARS-CoV-2. It is used to manufacture mouth-nose-covering masks using the latest technology. Many masks made by obsolete mechanization are useless for filtering small aerosols, ranging in size from 10 nm to 10 $\mu$m, whereas face masks related to graphene can filter such ultra-fine particles with excellent success. Hence, graphene-enhanced face masks receive hopeful responses from the users. Several charismatic aspects, such as simplicity of use, auto sterilization, ultra filtration of aerosol particles, reusable type, quick charging process, hydrophobic nature, not suffocating while breathing, durability, and cost-effectiveness, have made it very popular among people who are being hunted by the contagious SARS-CoV-2 virus. However, the interaction of graphene nanoparticles with viable cells and biochemical is considered unsuitable and dangerous to the human body. Although the toxicity of graphene nanoparticles varies depending on the concentration, the number of layers, surface charge density, purity, exposure time, stability, and particle size, even a minor presence of this nanomaterial inside the body can lead to serious chronic diseases, such as cancer. Reports have also shown that its interaction with microphagous cells can weaken the immune system of our body. Hence, we believe that the use of graphene-enhanced face masks is unfriendly. Therefore, there should be a surplus of investigations and studies on the use of such materials before commercialization.

### Funding
The authors received no funding for this work.

### Competing Interests
The authors declare there are no competing interests.

### Author Contributions
- Siyanand Kumar Chaudhary conceived and designed the experiments, performed the experiments, analyzed the data, prepared figures and/or tables, authored or reviewed drafts of the paper, and approved the final draft.
- Nabina Chaudhary conceived and designed the experiments, analyzed the data, authored or reviewed drafts of the paper, and approved the final draft.
- Rahul Chaudhary conceived and designed the experiments, performed the experiments, analyzed the data, performed the computation work, authored or reviewed drafts of the paper, and approved the final draft.
- Narendra Kumar Chaudhary conceived and designed the experiments, performed the experiments, performed the computation work, prepared figures and/or tables, authored or reviewed drafts of the paper, and approved the final draft.

### Data Availability
This is a literature review.

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
