# Peer review of "(untitled)"

_PeerJ Materials Science, doi:10.7717/peerj-matsci.20_

## Round 0.1 · original submission · Major Revisions

Reviewer 1 has requested that you cite specific references. You may add them if you believe they are especially relevant. However, I do not expect you to include these citations, and if you do not include them, this will not influence my decision.

Reviewer 1 ·

Basic reporting

No comment

Experimental design

No comment

Validity of the findings

No comment

Additional comments

The review paper presented intends to demonstrate the prospects and challenges of graphene-based face masks in preventing COVID-19 Pandemic. This is a very timely review and significant study since highly infectious COVID-19 has become a global public health concern. Graphene-enhanced face masks have demonstrated some hopeful results due to the incredible properties of graphene. However, there are health risks associated with the use of this graphene-enhanced face masks. This review paper has systematically reviewed recently published papers in relevant area and presented scientifically. However, it would be good to make few minor amendments:

Comment 1: Line 217: Check if the figure number is accurate.

Comment 2: It would be great to include some other relevant papers such as: https://doi.org/10.1021/acsnano.0c05537, https://doi.org/10.1002/adfm.202107407, https://doi.org/10.1002/adsu.202100176

Reviewer 3 ·

Basic reporting

The topic is interesting and hot; however, the article needs significant modification.

Experimental design

no comments

Validity of the findings

no comments

Additional comments

1. How does the electrical and thermal conductivity of graphene reduce the virus transmission?

2. Techniques for the Incorporation of graphene into the polymer matrix should be highlighted. for example; electrospinning, 3D printing, coatings............etc

3. The possible environmental impacts associated with the use of graphene-based face masks should be covered.

4. Filtration process should be elaborated. Also, the benefits of using graphene in the face mask should be explained.

---

## Round 0.2 · accepted · Accept

Both of the original reviewers are satisfied with the revised manuscript and recommend acceptance.

Reviewer 1 ·

Basic reporting

No comment

Experimental design

No comment

Validity of the findings

No comment

Additional comments

The article meets the PeerJ criteria regarding basic reporting, study design and validity of findings. The article should be accepted as is.

Reviewer 3 ·

Basic reporting

no comment

Experimental design

no comment

Validity of the findings

no comment

Additional comments

The paper can be accepted.